# Foodborne Event Detection Based on Social Media Mining: A Systematic Review

**DOI:** 10.3390/foods14020239

**Published:** 2025-01-14

**Authors:** Silvano Salaris, Honoria Ocagli, Alessandra Casamento, Corrado Lanera, Dario Gregori

**Affiliations:** Unit of Biostatistics, Epidemiology and Public Health, Department of Cardiac, Thoracic, and Vascular Sciences, University of Padova, via Loredan, 18, 35121 Padova, Italy; silvano.salaris@studenti.unipd.it (S.S.); honoria.ocagli@unipd.it (H.O.); alessandra.casamento@studenti.unipd.it (A.C.)

**Keywords:** foodborne illness, systematic review, machine learning, social media

## Abstract

Foodborne illnesses represent a significant global health challenge, causing substantial morbidity and mortality. Conventional surveillance methods, such as laboratory-based reporting and physician notifications, often fail to enable early detection, prompting the exploration of innovative solutions. Social media platforms, combined with machine learning (ML), offer new opportunities for real-time monitoring and outbreak analysis. This systematic review evaluated the role of social networks in detecting and managing foodborne illnesses, particularly through the use of ML techniques to identify unreported events and enhance outbreak response. This review analyzed studies published up to December 2024 that utilized social media data and data mining to predict and prevent foodborne diseases. A comprehensive search was conducted across PubMed, EMBASE, CINAHL, Arxiv, Scopus, and Web of Science databases, excluding clinical trials, case reports, and reviews. Two independent reviewers screened studies using Covidence, with a third resolving conflicts. Study variables included social media platforms, ML techniques (shallow and deep learning), and model performance, with a risk of bias assessed using the PROBAST tool. The results highlighted Twitter and Yelp as primary data sources, with shallow learning models dominating the field. Many studies were identified as having high or unclear risk of bias. This review underscored the potential of social media and ML in foodborne disease surveillance and emphasizes the need for standardized methodologies and further exploration of deep learning models.

## 1. Introduction

Foodborne illnesses represent a significant and preventable public health challenge that affects millions of individuals worldwide every year [1]. The consequences of these diseases range from mild discomfort to severe health complications, including hospitalization and even death [2]. The latest statistics on foodborne diseases, worldwide and in Europe, highlight a significant health burden. Globally, nearly 1 in 10 people fall ill each year because of eating contaminated food, resulting in more than 420,000 deaths. Children under the age of 5 are disproportionately affected, accounting for 125,000 deaths annually [3]. The World Health Organization (WHO) also estimated that each year, unsafe food causes 600 million cases of foodborne diseases globally [4]. In Europe, this phenomenon is also of great concern. The European Food Safety Authority (EFSA) reported that, in 2021, there were 4005 foodborne outbreaks in the EU, representing a 29.8% increase compared to 2020 [5]. A more detailed report from EFSA and the European Center for Disease Prevention and Control (ECDC) noted that campylobacteriosis and salmonellosis were the most reported zoonoses (diseases transmitted from animals to humans) in 2021. Notably, more foodborne outbreaks and cases were documented in 2021 than in 2020 in the EU. Salmonella Enteritidis is the most frequently reported in these outbreaks [6].

In the United States, the Centers for Disease Control and Prevention (CDC) estimates that foodborne diseases make 48 million people fall ill, hospitalize 128,000, and cause 3000 deaths each year [7]. In 2020, 299 outbreaks of foodborne diseases were reported, causing 5987 illnesses, 641 hospitalizations, and 14 deaths [8]. These statistics underscore the ongoing challenges in ensuring food safety and reducing the incidence of foodborne diseases worldwide. Traditional methods of detecting and responding to foodborne illnesses often rely on formal reporting systems, laboratory tests, and periodic inspections, which may be limited in scope, timeliness, and effectiveness [9,10,11]. Traditional methods of detecting and responding to foodborne illnesses offer advantages over social media data, including greater reliability, validated laboratory-confirmed diagnoses, standardized reporting systems, and higher accuracy, as they are less prone to misinformation and bias often found in unverified social media sources [11].

In recent years, integrating digital platforms, social networks, and sophisticated computational methods has revolutionized public health monitoring and management in various areas [12], not just tracking foodborne illnesses. In disease surveillance, social networks, mobile applications, and online platforms have significantly impacted the real-time monitoring and follow-up of COVID-19 cases [13]. They facilitate rapid dissemination of information, enabling authorities and the public to respond quickly to evolving situations [14]. Although the Google Flu Trends platform no longer exists, it attempted to use search query data to monitor flu activity, showcasing the potential for digital surveillance [15]. AI has been used to predict infectious disease dynamics and the effects of interventions. By analyzing data trends, AI can predict how diseases might spread under different circumstances, helping to prepare and plan for outbreaks [16]. While social media and AI-driven approaches offer significant potential for detecting foodborne illnesses, they also come with notable disadvantages. Data derived from social networks can be unreliable and prone to misinformation, as posts may reflect personal anecdotes, rumors, or panic rather than verified cases. A supposed outbreak reported on social media may be exaggerated or entirely fabricated, making it difficult to distinguish real events from false alarms [17,18].

These novel approaches can revolutionize the detection of outbreaks, understanding of public responses, and implementation of preventive measures [19].

This systematic review aimed to explore and synthesize the current literature on using ML and other innovative technologies to detect and manage foodborne illnesses using social media or online review platforms, where users can leave a star rating and a detailed review of a business. In particular, this review aimed to assess the effectiveness of social media analysis by investigating how platforms like Twitter and Yelp are used to identify unreported foodborne events, extract relevant information, and analyze potential outbreaks. It also examined the development and application of machine learning models to perform real-time detection, prediction, and response to foodborne illness incidents. It analyzes various online tools, algorithms, and frameworks to enhance traditional surveillance systems, including graph neural networks (GNNs), text-mining (TM), and natural language processing (NLP). The work can hint at how the public reacts to foodborne outbreaks, including their concerns, behaviors, and interactions on social media platforms.

This study aimed to explore the various methods for predicting foodborne diseases using social media data.

## 2. Materials and Methods

Inclusion criteria encompass all studies dedicated to preventing foodborne diseases through social networks, focusing on those that use data mining techniques for preventive measures. On the contrary, clinical trials, case reports, case series, and reviews fall under exclusion criteria. Furthermore, exclusions extend to studies that do not leverage social media data or employ data mining techniques.

### 2.1. Information Sources and Search Strategy

MEDLINE was searched through PubMed, EMBASE, CINAHL, Web of Science, Arxiv, and Scopus. There was no other restriction in the search string or date. The last search was performed on 15 December 2024.

### 2.2. Selection Process

Two independent reviewers screened the retrieved articles using Covidence systematic review software, Veritas Health Innovation, Melbourne, Australia (available at www.covidence.org) [20]. Discrepancies were resolved by consensus or consultation with a third reviewer.

### 2.3. Data Extraction

Article characteristics are presented as means or percentages based on variable types. The following variables were collected during the research process: author, year, type of study, social media platform, country in which the study was conducted, language analyzed, structure involved, purpose of the survey, results, use of machine learning (ML), ML models tested, best ML model, best performance, and value (%). These parameters were systematically collected to evaluate and analyze the studies considered. We employed a dual categorization to evaluate the use of machine learning algorithms, dividing them into ‘shallow’ and ‘deep’ learning methods. Shallow learning, often referred to as traditional ML, involves algorithms that function with minimal layers of computation. These methods are characterized by their simplicity and efficiency in handling structured and rectangular data. Shallow learning algorithms typically include logistic regression, support vector machines, decision trees, and k-nearest neighbors. In contrast, deep learning (DL) consists of a subset of machine learning techniques (MLTs) that involve complex neural networks with multiple layers of processing and abstraction. DL methods, such as convolutional neural networks (CNNs) and recurrent neural networks (RNNs), are known for their proficiency in handling unstructured data such as images, sound, raw signals, and text.

### 2.4. Assessment of Risk of Bias

The Prediction Model Risk of Bias (RoB) Assessment Tool (PROBAST) was employed to determine the RoB in the included studies. PROBAST is designed to assess RoB in diagnostic and prognostic prediction model studies and comprises four domains—participants, predictors, outcomes, and analysis—covering 20 signaling questions (Qs) [21]. A domain was judged to have a “low RoB” if all signaling questions were answered as ‘Yes’ (Y) or ‘Probably Yes’ (PY). On the other hand, a response of ‘No’ (N) or ‘Probably No’ (PN) to one or more questions within a domain denoted a potential for bias. A response of ‘No Information’ (NI) indicated insufficient information to assess the risk of bias in that domain. The PROBAST evaluation was conducted independently by two reviewers (AC and SS), and disagreements were resolved through discussion or consultation with a third reviewer, as necessary.

## 3. Results

### 3.1. Characteristics of the Studies Included

This systematic review covered studies conducted over a decade, from 2012 to 2024. Appendix A reports the PRISMA flowchart of the article screening process. Table 1 reports the list and main characteristics of the articles included in the review. Articles are the preferred type of communication, comprising 76.92% of publications (24 out of 31) [22,23,24,25], 27 (p. 201), [26,27,28,29,30,31,32,33,34,35,36,37,38,39,40,41,42,43,44,45].

In terms of the platforms used to conduct research, Twitter is the most explored, being the focal point in 1.94% of the studies (13 out of 31) [22,25,31,35,38,39,44,45,46,47,50,51,52]. Yelp also attracts attention, being crucial in 22.58% of research efforts (7 out of 31) [23,26,27,30,33,48,49]. Other platforms, such as Weibo and Facebook, represent a smaller portion of the studies, with Weibo being a key component in 9.67% of the investigations (3 out of 31) [28,32,36], Facebook being investigated in 6.45% of the studies (2 out of 31) [37,40], and feedback data from agencies like the Singapore Food Agency’s CRMS contributing to 3.22% of studies (1 out of 31) [41].

Geographically, the USA stands out as the main focus, with 23 studies conducted there [23,24,25,26,27,29,30,31,33,34,35,38,39,40,42,43,44,46,48,49,50,51]. The European Union (EU) and China are also subjects of exploration, with the EU representing 9.67% of the studies (3 out of 31) [22,28,32] and China accounting for 6.45% (2 out of 31) [36,37]. Additionally, Singapore and Australia contributed new geographic diversity with one study each (3.22%) [41,45].

In terms of linguistic focus, English emerged as the dominant language analyzed, enveloping a considerable 90.2% of the studies (26 out of 31) [23,24,25,26,27,29,30,31,32,33,34,35,38,39,40,41,42,43,44,45,46,47,48,49,50,51,52]. Other languages or combinations, such as English and German or exclusively Chinese and Italian, are notably less prevalent, each contributing to 3.23%% of the studies (1 out of 31) [22,32,37].

### 3.2. Characteristics of the Settings Considered

Examining the public settings involved in the food chain analyzed through social media platforms, restaurants emerged as a prominent point of interest, being integral in 90.48% of the studies (19 out of 31) [22,32,37].

Alternatives and combinations of other structures, such as homes, supermarkets, mass gatherings, trips, soccer games, and online retail platforms, are explored sporadically, each representing approximately 4.76% of the studies (1 out of 21) [17,20,29]. All these data are summarized in Table 1.

### 3.3. Keywords Used

Analyzing the 31 studies, 19 (61.29%) [23,25,26,27,30,32,33,34,35,36,37,38,39,42,43,44,45,47,48] reported keywords used to filter data related to food safety and health (Table 2).

Keywords such as “food poisoning” and “vomit” were recurrent, appearing in seven instances [27,30,33,38,39,43,44,47,48]. Other frequently used terms include “sick”, “illness”, “stomachache”, “diarrhea”, “nausea”, and “puke”, as highlighted in these studies [42,43,44,45]. Hashtags such as # foodsecurity, # foodinsecure, and # foodequity were also notable in studies exploring food security and related themes [45].

Co-occurrence analysis revealed frequent keyword pairings, such as (“vomit”, and “diarrhoea”) [30,33,38,43,44,48,52], and (“sick”, “vomit”) [27,30,42,48]. Notably, terms like “food poisoning” and “illness” emerged prominently in studies, while additional keywords such as “stomachache,” “puke,” and “nausea” were also observed. Hashtags, including # foodsecurity, # foodinsecure, and # foodequity, featured significantly in studies exploring food security themes [45]. Methodological analysis showed varied approaches in keyword generation, ranging from using pre-defined lists [23,25,36,38,40] to computational methodologies, such as ML or frequency-based selections [31,33,34,44,52]. Some methods also incorporated manual validation by experts [38]. Some methods also incorporated manual validation by experts (Table 3).

### 3.4. Machine Learning Techniques Applied

Among the studies explored, a discernible inclination toward the utilization of ML is evident, with a total of 26 studies employing ML techniques [22,24,25,26,27,28,29,30,31,32,33,34,36,38,39,40,41,42,43,44,45,46,47,49,50,52], as opposed to 5 that opted not to [23,35,37,48,51]. Delving deeper into the ML approaches adopted, shallow learning models were more prevalently utilized, being featured in 16 studies [22,24,25,26,27,28,29,30,32,33,40,42,45,47,49,50], whereas DL models appeared in 9 studies [31,34,36,38,39,41,43,44,52]. On the deep learning front, techniques like BERT, DistilBERT, XLNet [41], and EGAL [44] were notable. The authors of [43] introduced a novel approach combining pretrained BERT models (BERTweet, RoBERTa, BiLSTM, and MGADE), while latent Dirichlet allocation (LDA) was applied for topic modeling [43]. Support vector machines (SVMs) emerged as the most frequently utilized model, being the approach of choice in five studies [22,24,28,40,50]. This perhaps underscores its renowned versatility and efficacy in tackling various classification problems. An overview of performance metrics reveals a notable divergence in reporting practices across studies. F1 Score, a harmonic mean of precision and recall, surfaced as the most frequently reported metric, being cited in eight studies [24,29,30,31,34,38,39,49]. This suggests a tendency to evaluate models based on their ability to balance type I and type II errors once a threshold was selected, especially pertinent in imbalanced datasets. However, it is crucial to highlight the substantial variability in the performance metrics reported. Metrics like accuracy and AUC (area under curve) illustrate a broader range of values, ranging from 66.4% [47] to 92.0% [49] and 93.0% [26] to 98.0% [30], respectively (Table 2).

### 3.5. Risk of Bias Assessment

The detailed RoB assessment for each included study is compiled in Table 4 and Appendix A. Overall, as depicted in Table 4, Domain 1—Participants—saw a significant proportion of studies (15) categorized as having a “High RoB”, while a substantial number (15) remained ambiguous, falling under the “Unclear RoB” category. In Domain 2 Predictors, because Q2 was predominantly (77%) characterized by PN responses, the majority (20) of the studies were identified with a “High RoB”, with only one research, was considered to have a “Low RoB”. A similar assessment was reported in Domain 3—Outcome—with 21 instances marked as “High Risk”. Importantly, Domain 4—Analysis—was evaluated as “Unclear RoB” in 23 scrutinized publications. In this case, a prevalent trend of NI responses was observed in multiple questions.

## 4. Discussion

This systematic review explored the effectiveness of social networks and ML in detecting foodborne diseases. This review, which spans a decade of studies, demonstrated the central role of platforms such as Twitter and Yelp in using user-generated content for epidemiological research, with a significant focus on the English language and the US region. This could indicate various factors, such as data availability, technological infrastructure, or specific socio-cultural phenomena. Comparatively, those conducted by [53] also underscored the importance of social media in tracking infectious diseases or health trends. This similarity reinforces the potential of social networks as a valuable tool in public health monitoring and disease surveillance. It is important to emphasize that this approach serves as a complementary tool rather than a replacement for traditional surveillance methods. When used with great care, rigorously validated, and integrated with assessments by health authorities, social networks and machine learning can significantly enhance public health monitoring and improve the detection and response to foodborne disease outbreaks.

Despite the rapid advances in DL, our review indicated a noticeable preference for shallow ML models. Shallow learning models, while efficient and interpretable, may lack the capacity to analyze highly complex and unstructured data such as noisy social media posts [54]. This preference may arise from the desire for simpler, more interpretable models, as seen in previous studies [55]. Deep learning models (e.g., BERT and CNN) showed promising performance but remained underutilized due to computational challenges and dataset limitations. This is consistent with the findings of [56], where dataset limitations influenced model choice. This presents an exciting opportunity for further research, providing more profound insights into the prevailing practices and tendencies in applying ML in various studies.

Performance metrics were inconsistently reported across studies, with significant variability (e.g., accuracy ranging from 66.4% to 98%). F1 Score was the most frequently reported metric but was not universally applied. This inconsistency hinders cross-study comparisons and meta-analysis. A standardized approach for reporting machine learning performance metrics, including precision, recall, and F1 Score, is essential to ensure comparability. The variability may also reflect imbalanced datasets, where certain outcomes (e.g., disease vs. healthy) are over-represented, potentially leading to biased results.

The last couple of years have seen an increase in the application of Large Language Models (LLMs) for disease outbreaks. However, the use for detecting foodborne illnesses on social media has not been identified. For instance, it has been demonstrated that GPT prompting can effectively analyze social media posts to detect potential conjunctivitis outbreaks with accuracy comparable to human analysis [57], and also help determine the severity and prevalence of COVID-19 [58]. Moreover, GPT models and their open-source counterparts can be leveraged to quickly and efficiently understand public sentiments, such as those surrounding online discussions about vaccination [59]. Therefore, LLMs appear to be valuable tools for public health monitoring.

In this systematic review, the retrieved studies predominantly focus on the USA (23 studies), with far fewer studies conducted in Asia (e.g., China, India, and Japan) or other regions such as Europe and Australia. Surprisingly, specific studies detailing the use of social media platforms for this purpose in countries like India and Japan were not identified. This gap is notable given the significant application of digital technologies in the public health sectors of these countries. For instance, in India, modern technologies such as the internet and mobile phones are critical in bridging accessibility gaps in public health systems, particularly in rural areas facing significant health equity challenges. Similarly, Japan’s Society 5.0 initiative provides a robust framework for integrating advanced technologies like artificial intelligence, big data, and cloud computing to address various societal challenges, including public health [60]. Interestingly, we only identified one study set in Australia [45]. Nonetheless, in Australia, the past decade has witnessed a significant transformation in the healthcare sector with a rapid shift towards virtual healthcare services, illustrating a broader trend of digital adoption in public health [61]. This gap highlights a potential area for further research into how social media could be utilized for foodborne disease surveillance in these countries, building on their existing digital health infrastructure.

Our findings indicated that Twitter and Yelp dominate the current body of literature, collectively accounting for approximately 66% of the studies reviewed. This predominance is likely due to the public availability of data on these platforms, the structured nature of Yelp reviews, and the extensive use of Twitter for real-time communication. However, the future of academic engagement on Twitter appears uncertain, particularly following the shutdown of the academic research API, which significantly hindered scholars’ ability to access and analyze data on the platform. This change, along with Musk’s unscientific use of Twitter’s “polls” feature, his promotion of conspiracy theories about US elected officials, and his puerile demeanor, collectively contributed to an environment that many academics found increasingly unpalatable, leading them to either quit Twitter altogether or reduce their engagement with the platform [62].

In addition, this focus on a limited number of platforms may introduce a source bias, potentially overlooking valuable insights from alternative platforms, such as Facebook, Reddit, Instagram, or region-specific digital tools. It is important to recognize that different platforms may capture unique user behaviors and reporting patterns, particularly in under-represented regions. Therefore, future research should expand the scope of analysis to include diverse data sources, integrating alternative social media platforms and other digital tools to ensure broader and more equitable foodborne disease surveillance.

In recent years, real-time data acquisition technologies, such as those facilitated by the Internet of Things (IoT), have emerged as a complementary approach to social media-based foodborne disease detection. IoT devices in food factories and along supply chains allow for the continuous monitoring of critical parameters, such as temperature, humidity, and contamination levels, enabling the early detection of potential risks at the source [63]. While our review focused on user-generated content from social media platforms, the integration of real-time IoT data with social media mining could significantly enhance surveillance systems. This combined approach holds promise for providing a more comprehensive and immediate response to food safety threats, particularly in production environments where contamination risks are high. Future studies should explore the synergy between IoT-based monitoring and machine learning models applied to digital platforms to advance the field of foodborne disease surveillance.

Most studies (84.2%) focused on restaurants as the primary setting for foodborne disease surveillance. Other settings like homes, supermarkets, or mass gatherings were rarely considered. The heavy focus on restaurants may overlook other critical environments where foodborne diseases originate, such as households, schools, or supply chains.

### Limitations

This systematic review indicated a conspicuous sparsity in reporting specific performance metrics in all studies. This hampers the comprehensive evaluation of ML models and poses challenges in conducting meta-analyses, which could be crucial in synthesizing findings and forging advancements in the field. This highlights a potential opportunity for improvement in future research, with the implementation of a standardized protocol for reporting various performance metrics, enhancing the comprehensiveness and comparability of studies within the field, as suggested by [36].

Our comprehensive assessment using the PROBAST tool also revealed significant concerns about the risk of bias in the 26 studies reviewed. Most studies were evaluated as having a high RoB for three (Participants, Predictors, Outcome) out of four domains. This aligns with the findings of [46], suggesting that high RoB is common in published clinical prediction models and is associated with poor discriminative performance. The last domain, that is, analysis, was evaluated as an unclear RoB for most studies. The limited representation of “Low RoB” evaluations, particularly for Domains 1, 3, and 4, may underline the need for more rigorous methodologies in future research.

The current focus on a limited number of platforms, such as Twitter and Yelp, may introduce source bias and overlook valuable insights from alternatives like Facebook, Reddit, Instagram, or region-specific tools. Expanding the scope to include diverse data sources, as well as integrating real-time IoT-based monitoring systems, could significantly improve foodborne disease surveillance.

Finally, the predominant emphasis on restaurant settings suggests a gap in addressing other critical environments where foodborne illnesses may originate, such as homes, schools, supermarkets, and supply chains. Addressing these limitations through interdisciplinary approaches that combine advanced ML methods, diverse data streams, and rigorous validation will be essential for advancing foodborne disease surveillance and enhancing public health outcomes.

## 5. Conclusions

Summarizing a decade of research highlights the utility of social media platforms, particularly Twitter and Yelp, for improving epidemiological research and public health surveillance of foodborne events. ML models, in particular shallow models, have been instrumental in deriving insights from user-generated content. However, future research efforts could benefit significantly from adopting standardized reporting protocols for ML model performance and exploring the potential of DL models in contexts where sufficient data is available. Ensuring that methods evolve as technology advances will be critical to maintaining the effectiveness of social media mining in detecting food-related events.

## Figures and Tables

**Table 1 foods-14-00239-t001:** Overview of research publications analyzing social media platforms across various countries and languages from 2012 to 2024 and involved structures.

Author	Type	Social Media Platform	Country	Analyzed Language	Involved Structure	Aim of the Study	Results
Cui et al., 2017 [28]	Article	Weibo	China		Restaurant	Extract data from Weibo (Chinese Twitter) and develop an algorithm for foodborne disease event detection.	Built an SVM classifier to classify each tweet into positive class (foodborne disease case) and negative class (noise)
Denecke et al., 2013 [22]	Article	Twitter, Blogs, Forums, TV, Radio, Online News	EU	English, German	Home, Mass Gathering, Trips, Soccer Game	Study the usefulness of the M-Eco medical system for supporting and monitoring a population’s health status through social media.	A system can reduce overwhelming information to a manageable amount of signals. Experiments yielded a proportion between 5 and 20% of signals regarded as ‘relevant’ by the users; signals were mainly generated from Twitter.
Effland et al., 2018 [30]	Article	Yelp	USA	English	Restaurant	Development and evaluation of a system to identify mentions of foodborne illness in Yelp restaurant reviews.	Effective information extraction from social media sites. The DOHMH system for foodborne illness surveillance in online restaurant reviews from Yelp was instrumental in identifying ten outbreaks and 8523 reports of foodborne illness associated with NYC restaurants since July 2012.
Erraguntla et al., 2019 [40]	Article	Facebook, Twitter, ProMED Mail, World Health Organization (WHO), BBC Health News, CDC Morbidity and Mortality Weekly Reports, The Lancet Infectious Diseases, BMC Infectious Disease.	USA	English		Develop the Framework for Infectious Disease Analysis (FIDA), which integrates data and provides situational awareness, visualizations, predictions, and intervention assessments to extract health-related information for health situation awareness.	All the predictive models performed similarly, with SVM slightly better performance. SVM was chosen as the preferred predictive model.
Gao et al., 2021 [36]	Article	Weibo	China			Propose a novel GNN-based model for event detection in social media to evaluate the performance of EDGNN by comparing it with state-of-the-art baseline models over a real-world foodborne disease event dataset.	A new Event Detection model is proposed based on GNN (EDGNN), which showed effectiveness in experiments on a real-world dataset. The presented method does not rely on the unique features of the dataset so that it can be generalized for event detection on various social media platforms.
Glowacki et al., 2019 [35]	Article	Twitter	USA	English		Text-mining analysis to look at tweets from two foodborne Escherichia coli outbreaks	Demonstration that social media sites such as Twitter can be used as a tool by public health agencies that wish to identify concerns about foodborne disease outbreaks.
Harris et al., 2014 [46]	CDCP weekly report	Twitter	USA	English	Restaurant	On 23 March 2013, the Chicago Department of Public Health (CDPH) and its civic partners launched FoodBorne Chicago (6), a website to improve food safety in Chicago by identifying and responding to complaints on Twitter about possible foodborne illnesses.	The effectiveness of social media for foodborne illness surveillance suggests mining tweets and restaurant reviews might aid in identifying and taking action on localized foodborne illness complaints that would otherwise go unreported.
Harris et al., 2017 [47]	Report	Twitter	USA	English	Restaurant	Pilot study for evaluation of a Web-based Dashboard (HealthMap Foodborne Dashboard) to identify and respond to tweets About food poisoning from St Louis City residents.	The Web-based Dashboard captured 193 relevant tweets. Our replies to relevant tweets resulted in more filed reports than several previously existing foodborne illness reporting mechanisms in St Louis during the same time frame.
Harrison et al., 2014 [48]	CDC Weekly Report	Yelp	USA	English	Restaurant	Identify restaurant reviews on Yelp that refer to unreported foodborne illnesses.	The program identified 893 reviews that required further evaluation by a foodborne disease epidemiologist. Of the 893 reviews, 499 (56%) described an event consistent with foodborne illness, and 468 represented a disease within four weeks of the review or did not provide a period.
Hu et al., 2022 [38]	Article	Twitter	USA	English		Develop machine learning-based models for foodborne outbreak detection using Twitter’s publicly available annotated dataset, TWEET-FID.	The construction of TWEET-FID for multiple foodborne illness detection tasks, evaluation of single-task and multi-task models, and the observation of learning from weak labels.
Hu et al., 2023 [44]	Conference	Twitter	USA	English	Restaurants	Develop EGAL, a deep learning framework to detect foodborne illness from social media posts.	EGAL achieved 86.3% accuracy and 87.9% balanced accuracy (bACC).
Joaristi et al., 2016 [49]	Conference	Yelp	USA	English	Restaurant	Propose a new method to detect health-violating restaurants based on Yelp reviews and user behavior.	The proposed classification method improves the inspector’s ability and outperforms previous solutions.
Kate et al., 2014 [24]	Article	Internet Forums	USA	English		Apply text mining techniques to identify and mine food safety complaints posted by citizens on web data sources.	The platform is a valuable tool to monitor content related to food safety complaints on internet forums and can help improve food safety practices and standards.
Lee et al., 2023 [42]	Conference	Yelp	USA	English	Restaurants	Predict food safety violations using Yelp reviews to identify high-risk establishments.	SVM achieved 86% recall, identifying high-risk establishments for inspection.
Maharana et al., 2019 [34]	Article	Amazon reviews	USA	English	Online retail platform	Develop a framework for early identification of unsafe food products using Amazon product reviews and FDA recall data.	The approach can improve food safety by enabling early identification of unsafe foods, leading to timely recall and limiting the health and economic impact on the public.
Mejia et al., 2019 [33]	Article	Yelp	USA	English	Restaurant	Examine how online reviews of restaurants can be used to identify hygiene violations and provide insights into restaurants’ hygiene practices between inspections.	The study demonstrates the potential benefits of online review data in informing inspection strategies and outcomes. Social media content and machine learning can address persistent social issues like restaurant hygiene.
Molenaar et al., 2024 [45]	Article	Twitter	Australia	English	General Public	Use NLP tools for sentiment analysis and topic modeling to explore food security discussions.	VADER showed 0.478 coherence score; negative sentiment received the highest engagement.
Nsoesie et al., 2014 [23]	Article	Yelp	USA	English	Restaurant	Assess whether crowdsourcing via food service reviews can be used as a surveillance tool with the potential to support efforts by local public health departments.	Online illness reports from platforms like Yelp could complement traditional surveillance systems by providing near real-time information on foodborne diseases, implicated foods, and locations.
Rizzoli et al., 2021 [37]	Article	Facebook	EU	Italian		Explore how and to what extent knowledge and perceptions of food risks during pregnancy are shared on social networks (Facebook).	The main results show that food risk is not among the most discussed topics, and the least known and debated food risks are the most widespread (e.g., campylobacteriosis). Sometimes, food risks, when addressed, were minimized or denied, and the belief to be ‘less at risk’ than peers for such risk (i.e., optimistic bias)was observed.
Sadilek et al., [50]	Article	Twitter	USA	English	Restaurant	Development of nEmesis, which applies machine learning to real-time Twitter data and analyzes the text of these tweets to estimate the probability that the user suffers from foodborne illness.	The adaptive inspection process is 64% more effective at identifying problematic venues than the current state of the art.
Sadilek et al., 2018 [29]	Article	Google Search	USA	English	Restaurant	Build FINDER, a machine-learned model for real-time detection of foodborne illness using anonymous and aggregated web search and location data.	FINDER improves the accuracy of health inspections; restaurants identified by FINDER are 3.1 times as likely to be deemed unsafe during the inspection as restaurants identified by existing methods.
Schomberg et al., 2016 [26]	Article	Yelp	USA	English	Restaurant	Present a method able to form robust predictions of health code violation prevalence, identify restaurants with a high risk of health code violation, and validate increased surveillance coverage by using free text and tags created by Yelp reviewers.	The predictive model predicted health code violations in 78% of the restaurants receiving serious citations in our pilot study of 440 restaurants.
Serban et al., 2019 [31]	Article	Twitter	USA	English	Restaurant	Real-time processing of social media (Twitter) with SENTINEL: A syndromic surveillance system incorporating deep learning for classifying health-related tweets	The preliminary results are promising, with the system able to detect outbreaks of influenza-like illness symptoms, which existing official sources could then confirm. The Nowcasting module shows that using social media data can improve prediction for multiple diseases over simply using traditional data sources.
Tao et al., 2021 [39]	Article	Twitter	USA	English	Restaurant	Employ Twitter as the data source and modify the language model BERTweet not only to predict if a consumer’s post (a tweet) indicates an incidence of foodborne illness but also to extract critical entities related to the foodborne illness incidence automatically.	Trends in Twitter data can be indicative of real-world foodborne outbreaks. The dual-task BERTweet model effectively extracted food entities, but challenges remain due to the inherent noisiness of social media data.
Tao et al., 2023 [43]	Article	Twitter	USA	English	Restaurants	Develop a web-based tool for real-time foodborne illness outbreak detection using Twitter data.	Pretrained BERT models (BERTweet) showed robust performance for foodborne illness detection.
Tegtmeyer et al., 2012 [51]	Conference	Twitter	USA	English		Determine how to use social web tools to track, trace, and respond to foodborne illness using data streams, analytical tools, bots, and dashboards.	The primary focus of our work is on this dashboard system. The system will allow for different levels of participation from users, a curious browser, affected reported, and active reviewers.
Vasanthakumar et al., 2023 [41]	Conference	Other: feedback data from SFA’s CRMS	Singapore	English	Public feedback	Develop a method to automate the classification of urgency in food safety reports.	Fine-tuned BERT outperformed DistilBERT and XLNet for urgency classification tasks.
Wang et al., 2017 [27]	Article	Yelp	USA	English	Restaurant	Retrieve and analyze Yelp reviews to predict foodborne illnesses in restaurants to prevent more people from being affected.	After performing text classification, we used the models to predict whether the review indicates foodborne illness with high probabilities. SVM and RNN perform better than others, with higher accuracy and F-scores.
Widener et al., 2014 [25]	Article	Twitter	USA	English	Restaurant, Supermarket	Understand how geolocated tweets can be used to explore the prevalence of healthy and unhealthy food across the contiguous United States. Examine whether tweets about unhealthy foods are more common in these areas.	The results show that these disadvantaged census tracts tend to have a lower proportion of tweets about healthy foods with a positive sentiment and a higher proportion of unhealthy tweets in general. These findings substantiate the methods used by the USDA to identify regions that are at risk of having low access to healthy foods.
Zhang et al., 2019 [32]	Article	Weibo	China	Chinese	Restaurant	Improve the temporal and spatial precision of foodborne disease prediction based on big data.	The results determined the scientific issues regarding how to improve the temporal and spatial accuracy of foodborne disease outbreak risk prediction in Beijing.
Zou et al., 2016 [52]	Conference	Twitter	EU	English	Restaurant	Infer IID (infectious intestinal diseases) occurrences from Twitter in England.	Our experimental results regarding predictive performance and semantic interpretation indicate that Twitter data contain a signal that could be strong enough to complement conventional methods for IID surveillance.

**Table 2 foods-14-00239-t002:** List of keywords found in the analyzed papers, grouped by categories.

Symptoms	Bacteria/Viruses	General Terms
puke, diarrhea, nausea, Vomit *, Throw * up, Tummy upset, Tummy pain, The runs, vomiting, stomach ache, Rat-bite fever, hiccups, wiping nose, vomit, Puk *, Abdominal ache, Abdominal pain, heartburn, # stomachache, Stomach pain, indigestion	*Escherichia*, Giardia, Listeri *, Norovirus, *E.coli/Ecoli/E coli*, Rotavirus, *Salmonell* *, *Shigell* * *Staphylococcus, Helicobacter*, shiga toxin-producing *E. coli*, o157:h7, ehec, enterohemorrhagic *E. coli* serotype o157:h7, *Cyclospora*, *Cronobacter*	stench, employees, humid, septic, Jesus, hell, dishes, the_best, high_quality, adorable, fabulous, craving, favorite, excellent, service, recommend, professional, delicious, wash_hands, burnt, ache, pain, cigarette, asshole, awful, rotten, bathroom, toilet, fuck, microwaved, shit, bitch, sucks, mold, mice, spider, exclaim, filthy, roach, DIRTY, I_found_a, clean, food_poisoning, dirty, truck + sick, stomach + hospital, fish, terrible, horrible, Sick, Food poisoning, Cryptococcosis, Spit *, Heartburn, Acidosis, Gastro *, Cryptosporidiosis, Diarrhea, Amebiasis, Anisakiasis, Ascariasis, Anthrax, Botulism, Brucellosis, Campylobacteriosis, Ciguatoxin, Cysticercosis, Hepatitis A, Roundworm, Diphyllobothr *, Isosporiasis, Leptospirosis, Toxoplasmosis, Viral, food poisoning, # foodpoisoning, I_love, healthy foods, vomiting, Pregnancy, negative and hygiene-related words, prevention, FDA, food reservoirs, food safety, foodborne illness, foodborne transmission, hemolytic uremic syndrome, hus, out-break, phac, Public Health Agency of Canada, romaine lettuce, shiga toxin, stec, United States Food and Drug Administration, flu, cold, pesticide, Christ, Centers for Disease Control, affordable, reflux, CDC, stomach, rotten, pungency, foul, pool, poison, Being mothers in…., bacterium, ill, food poison, pushy, Weaning, bacteria, foodpoisoning, label, oldschool, Mothers, advisory

The asterisk (*) is used to capture all possible words sharing the same root (e.g., “vomit”, “vomiting”, “vomited”). The hashtag (#) is used on Twitter to categorize tweets or emphasize specific topics.

**Table 3 foods-14-00239-t003:** Evaluation of machine learning models in the analyzed studies, highlighting performance metrics and best achieved results.

Author (Year)	Use of Machine Learning	Tested ML Models	Best ML Model	Best Performance	Value (%)
Cui et al., 2017 [28]	Yes, shallow	SVM	SVM		
Denecke et al., 2013 [22]	Yes, shallow	SVM, K-means algorithm	SVM	Precision	92
Effland et al., 2018 [30]	Yes, shallow	J4.810 decision tree, Logistic Regression, RF, SVM	Logistic Regression	Precision	96
Erraguntla et al., 2019 [40]	Yes, shallow	SVM, Linear Regression (with nonlinear and interaction terms), decision tree-based boosting	SVM	Average Mean Square Error	14
Gao et al., 2021 [36]	Yes, deep	GNN, EDGNN, BiGRU, BERT, CRFTM, CNN, LSTM			
Harris et al., 2014 [46]	Yes	Supervised Learning Algorithm	Supervised Learning Algorithm		
Harris et al., 2017 [47]	Yes, shallow			Accuracy	66.4
Hu et al., 2022 [38]	Yes, deep	RoBERTa, BiLSTM, MGADE, MV, BSC,	RoBERTa	Accuracy	84.7
Hu et al., 2023 [44]	Yes, deep	EGAL	EGAL	Accuracy	86.3
Joaristi et al., 2016 [49]	Yes, shallow	SVM, Logistic Regression, and Random Forest		F1 Score Value	95
Kate et al., 2014 [24]	Yes, shallow	Multinomial Naive Bayes, k-NN, SVM	SVM	Recall	67.3
Lee et al., 2023 [42]	Yes, shallow	Decision Tree, Random Forest, SVM	SVM	Recall	86
Maharana et al., 2019 [34]	Yes, deep	SVM, Multinomial Naive Bayes, Weighted Logistic Regression, BERT, Autoencoder Neural Network	BERT	Precision	78
Mejia et al., 2019 [33]	Yes, shallow	Naive Bayes classifier, Linear Regressions			
Molenaar et al., 2024 [45]	Yes, shallow	VADER (Sentiment Analysis), Latent Dirichlet Allocation (LDA)	VADER	Coherence Score	0.47
Sadilek et al., [50]	Yes, shallow	SVM	SVM	Recall	96
Sadilek et al., 2018 [29]	Yes, shallow	Supervised Machine-learned Classifier		Roc	85
Schomberg et al., 2016 [26]	Yes, shallow	Logistic Regression,	Logistic Regression	Auc	98
Serban et al., 2019 [31]	Yes, deep	CNN, SVM, Multinomial Naive Bayes	CNN	Accuracy	85.4
Tao et al., 2021 [39]	Yes, deep	Dual-task BERTweet	Dual-task BERTweet model	Recall	88.6
Tao et al., 2023 [43]	Yes, deep	BERTweet, RoBERTa, BiLSTM, MGADE, LDA	BERTweet	Balanced Accuracy (bACC)	87.9
Vasanthakumar et al., 2023 [41]	Yes, deep	BERT, DistilBERT, XLNet	BERT	Precision	86
Wang et al., 2017 [27]	Yes, shallow	NB, SVM, RF, RNN, generalized linear model			
Widener et al., 2014 [25]	Yes, shallow	Logistic Regression	Logistic Regression		
Zhang et al., 2019 [32]	Yes, shallow	Bayesian Regression, Linear Regression, ElasticNet Regression, SVR, and GBR			
Zou et al., 2016 [52]	Yes, deep	Elastic Net, GP, Skip-gram for word embeddings		Pearson Correlation (R)	71.1

Abbreviations: SVM = support vector machine, RF = random forest, GNN = graph neural network, EDGNN = event detection model based on graph neural network, BiGRU = bidirectional gated recurrent unit, BERT = bidirectional encoder representations from transformers, CRFTM = conditional random field topic model, CNN = convolutional neural network, LSTM = long short-term memory, RoBERTa = robustly optimized BERT pretraining approach, BiLSTM = bidirectional long short-term memory, MGADE = multi-grained attention with denoising encoder, MV = majority voting, BSC = Bayesian sequence combination, NB = naive bayes, RNN = recurrent neural network, SVR = support vector regression, GBR = gradient boosting regression, GP = gaussian process.

**Table 4 foods-14-00239-t004:** Overall judgment of risk of bias for each domain.

Author (Year)	Domain 1: Participants	Domain 2: Predictors	Domain 3: Outcome	Domain 4: Analysis
Cui et al., 2017 [28]	Unclear Risk	Unclear Risk	Unclear Risk	Unclear Risk
Denecke et al., 2013 [22]	High Risk	High Risk	High Risk	Unclear Risk
Effland et al., 2018 [30]	High Risk	High Risk	High Risk	Unclear Risk
Erraguntla et al., 2019 [40]	High Risk	High Risk	High Risk	Unclear Risk
Gao et al., 2021 [36]	High Risk	High Risk	High Risk	Unclear Risk
Glowacki et al., 2019 [35]	Unclear Risk	Unclear Risk	Unclear Risk	Unclear Risk
Harris et al., 2014 [46]	Unclear Risk	Unclear Risk	Unclear Risk	Unclear Risk
Harris et al., 2017 [47]	High Risk	High Risk	High Risk	Unclear Risk
Harrison et al., 2014 [48]	Unclear Risk	Low Risk	Unclear Risk	Unclear Risk
Hu et al., 2022 [38]	High Risk	High Risk	High Risk	Unclear Risk
Hu et al., 2023 [44]	Low Risk	Unclear Risk	High Risk	High Risk
Joaristi et al., 2016 [49]	High Risk	High Risk	High Risk	High Risk
Kate et al., 2014 [24]	High Risk	High Risk	High Risk	Unclear Risk
Lee et al., 2023 [42]	Unclear Risk	Unclear Risk	Unclear Risk	Unclear Risk
Maharana et al., 2019 [34]	High Risk	High Risk	High Risk	High Risk
Mejia et al., 2019 [33]	High Risk	High Risk	High Risk	Unclear Risk
Molenaar et al., 2024 [45]	Unclear Risk	Unclear Risk	Unclear Risk	Unclear Risk
Nsoesie et al., 2014 [23]	High Risk	High Risk	High Risk	High Risk
Rizzoli et al., 2021 [37]	High Risk	High Risk	High Risk	High Risk
Sadilek et al., [50]	Unclear Risk	High Risk	High Risk	Unclear Risk
Sadilek et al., 2018 [29]	Unclear Risk	High Risk	High Risk	Unclear Risk
Schomberg et al., 2016 [26]	Unclear Risk	Unclear Risk	Unclear Risk	Unclear Risk
Serban et al., 2019 [31]	High Risk	High Risk	High Risk	High Risk
Tao et al., 2021 [39]	High Risk	High Risk	High Risk	High Risk
Tao et al., 2023 [43]	Unclear Risk	Unclear Risk	Unclear Risk	Unclear Risk
Tegtmeyer et al., 2012 [51]	Unclear Risk	Unclear Risk	Unclear Risk	Unclear Risk
Vasanthakumar et al., 2023 [41]	Unclear Risk	Unclear Risk	Unclear Risk	Unclear Risk
Wang et al., 2017 [27]	Unclear Risk	High Risk	High Risk	Unclear Risk
Widener et al., 2014 [25]	High Risk	High Risk	High Risk	High Risk
Zhang et al., 2019 [32]	Unclear Risk	High Risk	High Risk	Unclear Risk
Zou et al., 2016 [52]	Unclear Risk	High Risk	High Risk	Unclear Risk

## Data Availability

The original contributions presented in the study are included in the article/Appendix A, further inquiries can be directed to the corresponding authors.

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
