# Peer review of "Foodborne Event Detection Based on Social Media Mining: A Systematic Review"

_foods, 2025, doi:10.3390/foods14020239_

Round 1

Reviewer 1 Report

Comments and Suggestions for Authors

Foodborne disease transmission is one of the most relevant challenges for global health. In contrast, conventional surveillance methods only sometimes help in early detection. Therefore, this study proposes innovative solutions that incorporate social media platforms and learning. Systemic analysis is applied to the networks, using ML techniques that identify the presence of unreported events in order to develop improvement strategies. PROBAST evaluated the risk of bias, where cases of high or unclear bias were identified. Therefore, it is necessary to standardize the role of social networks in the role of foodborne disease surveillance. In the introduction, the current importance of the role of social networks is precisely stated. On the other hand, the methodology has technical support embodied in four points. The results include an extensive explanation of the role of these media in China and the USA. However, countries such as Japan, Australia, India, are not mentioned. The discussion needs to be more extensive in comparison with the data obtained in the results.

Author Response

Reviews SR food 

Open Review 

(x) I would not like to sign my review report 
( ) I would like to sign my review report 

Quality of English Language 

(x) The quality of English does not limit my understanding of the research. 
( ) The English could be improved to more clearly express the research. 

Yes 

Can be improved 

Must be improved 

Not applicable 

Does the introduction provide sufficient background and include all relevant references? 

(x) 

( ) 

( ) 

( ) 

Is the research design appropriate? 

(x) 

( ) 

( ) 

( ) 

Are the methods adequately described? 

(x) 

( ) 

( ) 

( ) 

Are the results clearly presented? 

(x) 

( ) 

( ) 

( ) 

Are the conclusions supported by the results? 

( ) 

(x) 

( ) 

( ) 

Comments and Suggestions for Authors 

 We thank the reviewer for their insightful and constructive comments, which have helped improve our manuscript. Below, we outline the actions taken in response to each suggestion. 

We hope these revisions address the reviewer’s concerns and further strengthen the manuscript. 

Foodborne disease transmission is one of the most relevant challenges for global health. In contrast, conventional surveillance methods only sometimes help in early detection. Therefore, this study proposes innovative solutions that incorporate social media platforms and learning. Systemic analysis is applied to the networks, using ML techniques that identify the presence of unreported events in order to develop improvement strategies. PROBAST evaluated the risk of bias, where cases of high or unclear bias were identified. Therefore, it is necessary to standardize the role of social networks in the role of foodborne disease surveillance. In the introduction, the current importance of the role of social networks is precisely stated. On the other hand, the methodology has technical support embodied in four points. The results include an extensive explanation of the role of these media in China and the USA. However, countries such as Japan, Australia, India, are not mentioned. The discussion needs to be more extensive in comparison with the data obtained in the results. 

  1. Inclusion of other countries (Japan, Australia, India): 
    We appreciate the reviewer highlighting the need to discuss additional countries. We have added a paragraph in the discussion.  
    In this systematic review, the retrieved studies predominantly focus on the USA (23 studies), with far fewer studies conducted in Asia (e.g., China, India, Japan) or other regions such as Europe and Australia. Surprisingly, specific studies detailing the use of social media platforms for this purpose in countries like India and Japan were not identified. This gap is notable given the significant application of digital technologies in the public health sectors of these countries. For instance, in India, modern technologies such as the internet and mobile phones are critical in bridging accessibility gaps in public health systems, particularly in rural areas facing significant health equity challenges. Similarly, Japan’s Society 5.0 initiative provides a robust framework for integrating advanced technologies like artificial intelligence, big data, and cloud computing to address various societal challenges, including public health [66]. Interestingly, we only identified one study set in Australia [52]. Nonetheless, in Australia, the past decade has witnessed a significant transformation in the healthcare sector with a rapid shift towards virtual healthcare services, illustrating a broader trend of digital adoption in public health [67]. This gap highlights a potential area for further research into how social media could be utilized for foodborne disease surveillance in these countries, building on their existing digital health infrastructure. " 
  1. Expanded discussion: 
    We have revised the discussion section to provide a broader comparison of our findings. Specifically, we have contextualized our results by comparing them to previous studies and highlighting regional differences where applicable.  

Despite the rapid advances in DL, our review indicates a noticeable preference for shallow ML models. Shallow learning models, while efficient and interpretable, may lack the capacity to analyze highly complex and unstructured data such as noisy social media posts [60]. This preference may arise from the desire for simpler, more interpretable models, as seen in studies [61]. Deep learning models (e.g., BERT, CNN) showed promising performance but remain underutilized due to computational challenges and dataset limitations. Alternatively, this might be due to dataset size limitations, critical in training more complex DL models. This is consistent with the findings of [62], where data set limitations influenced model choice. This presents an exciting opportunity for further research, providing more profound insights into the prevailing practices and tendencies in applying ML in various studies. 

Performance metrics were inconsistently reported across studies, with significant variability (e.g., accuracy ranging from 66.4% to 98%). F1 Score was the most frequently reported metric but was not universally applied. This inconsistency hinders cross-study comparisons and meta-analysis. A standardized approach for reporting machine learning performance metrics, including precision, recall, and F1 score, is essential to ensure comparability. The variability may also reflect imbalanced datasets, where certain outcomes (e.g., disease vs. healthy) are overrepresented, potentially leading to biased results. 

The last couple of years have seen an increase in the application of of Large Language Models (LLMs) for disease outbreaks. However the use for detecting foodborne illnesses on social media has not been identified. For instance, it has been demonstrated that GPT prompting can effectively analyze social media posts to detect potential conjunctivitis outbreaks with accuracy comparable to human analysis [63], and also help determine the severity and prevalence of COVID-19 [64]. Moreover, GPT models and their open-source counterparts can be leveraged to quickly and efficiently understand public sentiments, such as those surrounding online discussions about vaccination [65]. Therefore, LLMs appear to be valuable tools for public health monitoring. 

In this systematic review, the retrieved studies predominantly focus on the USA (23 studies), with far fewer studies conducted in Asia (e.g., China, India, Japan) or other regions such as Europe and Australia. Surprisingly, specific studies detailing the use of social media platforms for this purpose in countries like India and Japan were not identified. This gap is notable given the significant application of digital technologies in the public health sectors of these countries. For instance, in India, modern technologies such as the internet and mobile phones are critical in bridging accessibility gaps in public health systems, particularly in rural areas facing significant health equity challenges. Similarly, Japan’s Society 5.0 initiative provides a robust framework for integrating advanced technologies like artificial intelligence, big data, and cloud computing to address various societal challenges, including public health [66]. Interestingly, we only identified one study set in Australia [52]. Nonetheless, in Australia, the past decade has witnessed a significant transformation in the healthcare sector with a rapid shift towards virtual healthcare services, illustrating a broader trend of digital adoption in public health [67]. This gap highlights a potential area for further research into how social media could be utilized for foodborne disease surveillance in these countries, building on their existing digital health infrastructure. 

Our findings indicate that Twitter and Yelp dominate the current body of literature, collectively accounting for approximately 66% of the studies reviewed. This predominance is likely due to the public availability of data on these platforms, the structured nature of Yelp reviews, and the extensive use of Twitter for real-time communication. However, the future of academic engagement on Twitter appears uncertain, particularly following the shutdown of the academic research API, which significantly hindered scholars' ability to access and analyze data on the platform. This change, along with Musk's unscientific use of Twitter's "polls" feature, his promotion of conspiracy theories about U.S. elected officials, and his puerile demeanor, collectively contributed to an environment that many academics found increasingly unpalatable, leading them to either quit Twitter altogether or reduce their engagement with the platform [68]. 

In addition, this focus on a limited number of platforms may introduce a source bias, potentially overlooking valuable insights from alternative platforms, such as Facebook, Reddit, Instagram, or region-specific digital tools. It is important to recognize that different platforms may capture unique user behaviors and reporting patterns, particularly in underrepresented regions. Therefore, future research should expand the scope of analysis to include diverse data sources, integrating alternative social media platforms and other digital tools to ensure broader and more equitable foodborne disease surveillance. 

In recent years, real-time data acquisition technologies, such as those facilitated by the Internet of Things (IoT), have emerged as a complementary approach to social media-based foodborne disease detection. IoT devices in food factories and along supply chains allow for the continuous monitoring of critical parameters, such as temperature, humidity, and contamination levels, enabling early detection of potential risks at the source [69]. While our review focused on user-generated content from social media platforms, the integration of real-time IoT data with social media mining could significantly enhance surveillance systems. This combined approach holds promise for providing a more comprehensive and immediate response to food safety threats, particularly in production environments where contamination risks are high. Future studies should explore the synergy between IoT-based monitoring and machine learning models applied to digital platforms to advance the field of foodborne disease surveillance. 

Most studies (84.2%) focused on restaurants as the primary setting for foodborne disease surveillance. Other settings like homes, supermarkets, or mass gatherings were rarely considered. The heavy focus on restaurants may overlook other critical environments where foodborne diseases originate, such as households, schools, or supply chains.” 

Reviewer 2 Report

Comments and Suggestions for Authors

1. As a review paper, I think you have too few references. Usually, a good review paper should have more than 100 references.

2. Machine learning technology is updated very quickly. Your analysis is up to August 2022. You should cite more updated references, especially those from 2024.

3. In addition to the data sources you mentioned, it is recommended to consider relevant literature on public and private data sets, especially some real-time data, which is currently a mainstream technology, such as real-time data quickly obtained by the Internet of Things in food factories. And the two platforms you mentioned account for about 66% in total.

4. I think that in the past three years, there should be more applications of deep learning, but considering that the time range of your literature is from 2012 to 2022, there is nothing wrong with the shallow learning models you mentioned. But I still recommend that you use more recent references, which will change some of your findings.

5. This is my personal suggestion. Authors can consider it carefully according to the actual situation.

Author Response

20 November 2024 

Date of this review 

03 Dec 2024 01:31:52 

Open Review 

(x) I would not like to sign my review report 
( ) I would like to sign my review report 

Quality of English Language 

(x) The quality of English does not limit my understanding of the research. 
( ) The English could be improved to more clearly express the research. 

Is the work a significant contribution to the field? 

[Controllo][Controllo][Controllo][Controllo][Controllo] 

Is the work well organized and comprehensively described? 

[Controllo][Controllo][Controllo][Controllo][Controllo] 

Is the work scientifically sound and not misleading? 

[Controllo][Controllo][Controllo][Controllo][Controllo] 

Are there appropriate and adequate references to related and previous work? 

[Controllo][Controllo][Controllo][Controllo][Controllo] 

Is the English used correct and readable? 

[Controllo][Controllo][Controllo][Controllo][Controllo] 

Comments and Suggestions for Authors 

  1. As a review paper, I think you have too few references. Usually, a good review paper should have more than 100 references.
  2. Machine learning technology is updated very quickly. Your analysis is up to August 2022. You should cite more updated references, especially those from 2024.

We thank the reviewer for pointing this out. We have updated the review to include new results. However, in the field of foodborne disease detection, the models identified were largely similar to those already retrieved. Notably, we observed the emerging use of large language models for outbreak detection through social media, as seen during the COVID-19 pandemic.  

The last couple of years have seen an increase in the application of of Large Language Models (LLMs) for disease outbreaks. However the use for detecting foodborne illnesses on social media has not been identified. For instance, it has been demonstrated that GPT prompting can effectively analyze social media posts to detect potential conjunctivitis outbreaks with accuracy comparable to human analysis [63], and also help determine the severity and prevalence of COVID-19 [64]. Moreover, GPT models and their open-source counterparts can be leveraged to quickly and efficiently understand public sentiments, such as those surrounding online discussions about vaccination [65]. Therefore, LLMs appear to be valuable tools for public health monitoring. 

  1. In addition to the data sources you mentioned, it is recommended to consider relevant literature on public and private data sets, especially some real-time data, which is currently a mainstream technology, such as real-time data quickly obtained by the Internet of Things in food factories. And the two platforms you mentioned account for about 66% in total.

We thank the reviewer for their valuable suggestion. We acknowledge the importance of real-time data sources, such as those provided by the Internet of Things (IoT) in food production facilities, for monitoring food safety risks. While our review focused on social media platforms due to their role in capturing user-generated reports of foodborne events, we agree that IoT technology represents a complementary and emerging approach that warrants exploration in future studies. 

In recent years, real-time data acquisition technologies, such as those facilitated by the Internet of Things (IoT), have emerged as a complementary approach to social media-based foodborne disease detection. IoT devices in food factories and along supply chains allow for the continuous monitoring of critical parameters, such as temperature, humidity, and contamination levels, enabling early detection of potential risks at the source [69]. While our review focused on user-generated content from social media platforms, the integration of real-time IoT data with social media mining could significantly enhance surveillance systems. This combined approach holds promise for providing a more comprehensive and immediate response to food safety threats, particularly in production environments where contamination risks are high. Future studies should explore the synergy between IoT-based monitoring and machine learning models applied to digital platforms to advance the field of foodborne disease surveillance. 

Regarding the dominance of Twitter and Yelp, our analysis reflects the current body of literature, where these platforms are most frequently utilized due to their accessibility and availability of public data. However, we have now highlighted this as a limitation and suggested that future research should consider integrating data from additional platforms and real-time systems to provide a more comprehensive surveillance framework. 
Our findings indicate that Twitter and Yelp dominate the current body of literature, collectively accounting for approximately 66% of the studies reviewed. This predominance is likely due to the public availability of data on these platforms, the structured nature of Yelp reviews, and the extensive use of Twitter for real-time communication. However, the future of academic engagement on Twitter appears uncertain, particularly following the shutdown of the academic research API, which significantly hindered scholars' ability to access and analyze data on the platform. This change, along with Musk's unscientific use of Twitter's "polls" feature, his promotion of conspiracy theories about U.S. elected officials, and his puerile demeanor, collectively contributed to an environment that many academics found increasingly unpalatable, leading them to either quit Twitter altogether or reduce their engagement with the platform [68]. 

In addition, this focus on a limited number of platforms may introduce a source bias, potentially overlooking valuable insights from alternative platforms, such as Facebook, Reddit, Instagram, or region-specific digital tools. It is important to recognize that different platforms may capture unique user behaviors and reporting patterns, particularly in underrepresented regions. Therefore, future research should expand the scope of analysis to include diverse data sources, integrating alternative social media platforms and other digital tools to ensure broader and more equitable foodborne disease surveillance. 

  1. I think that in the past three years, there should be more applications of deep learning, but considering that the time range of your literature is from 2012 to 2022, there is nothing wrong with the shallow learning models you mentioned. But I still recommend that you use more recent references, which will change some of your findings.

Thanks, revised results and discussion accordingly. 

  1. This is my personal suggestion. Authors can consider it carefully according to the actual situation.

Submission Date 

20 November 2024 

Date of this review 

30 Nov 2024 09:05:01 

Reviewer 3 Report

Comments and Suggestions for Authors

Suggestions for corrections are in the attached file. The article has scientific merit and can be accepted after the requested corrections have been made.

Author Response

 We thank the reviewer for their insightful and constructive comments, which have helped improve our manuscript. Below, we outline the actions taken in response to each suggestion. 

We hope these revisions address the reviewer’s concerns and further strengthen the manuscript. 

Round 2

Reviewer 2 Report

Comments and Suggestions for Authors

The author made a lot of revisions and the quality of the manuscript has been greatly improved.